# MAKING PREDICTORS MORE RELIABLE WITH SELECTIVE RECALIBRATION

## ABSTRACT

A reliable deep learning system should be able to accurately express its confidence with respect to its predictions, a quality known as calibration. One of the most effective ways to produce reliable confidence estimates with a pre-trained model is by applying a post-hoc recalibration method. Popular recalibration methods like temperature scaling are typically fit on a small amount of data and work in the model's output space, as opposed to the more expressive feature embedding space, and thus usually have only one or a handful of parameters. However, the target distribution to which they are applied is often complex and difficult to fit well with such a function. To this end we propose *selective recalibration*, where a selection model learns to reject some user-chosen proportion of the data in order to allow the recalibrator to focus on regions of the input space that can be well-captured by such a model. We provide theoretical analysis to motivate our algorithm, and test our method through comprehensive experiments on difficult medical imaging and zero-shot classification tasks. Our results show that selective recalibration consistently leads to significantly lower calibration error than a wide range of selection and recalibration baselines.

## 1 INTRODUCTION

In order to build user trust in a machine learning system, it is important that the system can accurately express its confidence with respect to its own predictions. Under the notion of calibration common in deep learning (Guo et al., 2017; Minderer et al., 2021), a confidence estimate output by a model should be as close as possible to the expected accuracy of the prediction. For example, a prediction assigned 30% confidence should be correct 30% of the time. This is especially important in high-impact settings such as medical diagnosis, where a 30% chance of disease must be treated differently than 1% chance. While advancements in neural network architecture and training have brought improvements in calibration as compared to previous methods (Minderer et al., 2021), neural networks still suffer from miscalibration, usually in the form of overconfidence (Guo et al., 2017; Wang et al., 2021). In addition, these models are often applied to complex data distributions, possibly including outliers, and may have different calibration error within and between different subsets in the data (Ovadia et al., 2019; Perez-Lebel et al., 2023). We illustrate this setting in Figure 1(a) with a Reliability Diagram, a tool for visualizing calibration by plotting average confidence vs. accuracy for bins of datapoints with similar confidence estimates.

To address this calibration error, the confidence estimates of a pre-trained model can be refined using a post-hoc recalibration method like Platt scaling (Platt, 1999), temperature scaling (Guo et al., 2017), or histogram binning (Zadrozny & Elkan, 2001). While these "recalibrators" are effective, they are typically fit on small validation sets (on the order of hundreds to a few thousand examples), and also usually reduce the input space to the model's logits (e.g. temperature scaling) or predicted class scores (e.g. Platt scaling, histogram binning), as opposed to the high-dimensional and expressive feature embedding space. Accordingly, they are generally inexpressive models, having only one or a handful of parameters (Platt, 1999; Guo et al., 2017; Zadrozny & Elkan, 2001; Kumar et al., 2019). But the complex data distributions to which neural networks are often applied are difficult to fit well with such simple functions, and calibration error can even be exacerbated for some regions of the input space, especially when the model has only a single scaling parameter (see Figure 1(b)).

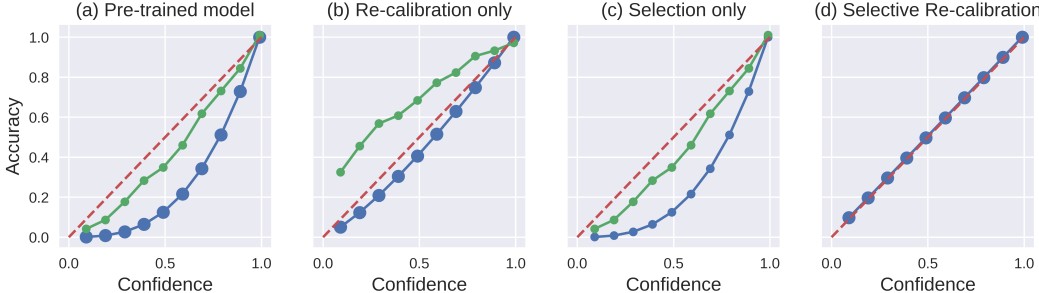

Figure 1: Reliability Diagrams for a model that has different calibration error (deviation from the diagonal) in different subsets of the data (here shown in blue and green). The size of each marker represents the amount of data in each bin, and the red dashed diagonal represents perfect calibration.

Motivated by these observations, we contend that these popular recalibration methods are a natural fit for use with a selection model. Selection models (El-Yaniv & Wiener, 2010; Geifman & El-Yaniv, 2017) are used alongside a classifier and may reject some portion of the classifier's predictions in order to improve performance on the subset of accepted (i.e. unrejected) examples. Selection models have been applied to the task of improving classifier accuracy (Geifman & El-Yaniv, 2017) or improving calibration error by rejecting the confidence estimates of a fixed model (Fisch et al., 2022). However, selection alone cannot address the underlying miscalibration because it does not alter the confidence output of the model (see Figure 1(c)), and the connection between selection and post-hoc recalibration remains largely unexplored.

In this work we propose *selective recalibration*, where a selection model and a recalibration model are jointly optimized in order to produce predictions with low calibration error. By rejecting some portion of the data, the system can focus on a region that can be well-captured by a simple recalibration model, leading to a set of predictions with a lower calibration error than under recalibration or selection alone (see Figure 1(d)). This approach is especially important when a machine learning model is deployed for decision-making in complex domains such as healthcare, lending, and legal systems, where the predictor must be well-calibrated in order that a human expert can use its output to improve outcomes and avoiding causing active harm. To summarize our contributions:

- We formulate selective recalibration, and offer a new loss function for training such a system, Selective Top-Label Binary Cross Entropy (S-TLBCE), which aligns with the typical notion of loss under smooth recalibrators like Platt or temperature scaling models.

- We test selective recalibration and S-TLBCE in real-world medical diagnosis and image classification experiments, and find that selective recalibration consistently leads to significantly lower calibration error than a wide range of selection and recalibration baselines.

- We provide theoretical insight to support our motivations and algorithm.

## 2 RELATED WORK

Making well-calibrated predictions is a key feature of a reliable statistical model (Guo et al., 2017; Hebert-Johnson et al., 2018; Minderer et al., 2021; Fisch et al., 2022). Popular methods for improving calibration error given a labeled validation dataset include Platt (Platt, 1999) and temperature scaling (Guo et al., 2017), histogram binning (Zadrozny & Elkan, 2001), and Platt binning (Kumar et al., 2019) (as well as others like those in Naeini et al. (2015); Zhang et al. (2020)). Calibration error is typically measured using quantities such as Expected Calibration Error (Naeini et al., 2015; Guo et al., 2017), Maximum Calibration Error (Naeini et al., 2015; Guo et al., 2017), or Brier Score (Brier, 1950) that measure whether prediction confidence matches expected outcomes. Another technique for improving ML system reliability is selective classification (Geifman & El-Yaniv, 2017; El-Yaniv & Wiener, 2010), wherein a model is given the option to abstain from making a prediction on certain examples (often based on confidence or out-of-distribution measures).

Recent work by Fisch et al. (2022) introduces the setting of calibrated selective classification, in which predictions from a pre-trained model are rejected for the sake of improving selective calibration error. The authors propose a method for training a selective calibration model using an S-MMCE loss function derived from the work of (Kumar et al., 2018). By considering the *joint* training and application of selection and re-calibration models, our work differs from this and other previous work. While Fisch et al. (2022) apply selection directly to a frozen model's outputs, We contend that the value in our algorithm lies in this joint optimization. Also, instead of using S-MMCE, we propose a new loss function, S-TLBCE, which more closely aligns with the objective function for Platt and temperature scaling.

## 3 BACKGROUND

Consider the multi-class classification setting with $K$ classes and data instances $(x, y) \sim \mathcal{D}$, where $x$ is the input and $y \in \{1, 2, ..., K\}$ is the ground truth class label. For a black box predictor $f$, $f(x) \in \mathbb{R}^K$ is a vector where $f(x)_k$ is the predicted probability that input $x$ has label $k$; we denote the confidence in the top predicted label as $\hat{f}(x) = \max_k f(x)_k$. Further, we may access the unnormalized class scores $f_s(x) \in \mathbb{R}^K$ (which may be negative) and the feature embeddings $f_e(x) \in \mathbb{R}^d$. The predicted class is $\hat{y} = \text{argmax}_k f(x)_k$ and the correctness of a prediction is $y^c = \mathbf{1}\{y = \hat{y}\}$.

**Selective Classification**. In selective classification, a selector $g$ produces binary outputs, where $0$ indicates rejection and $1$ indicates acceptance. A common goal is to decrease some error metric by rejecting no more than a $1 - \beta$ proportion of the data for given target coverage level $\beta$. One popular choice for input to $g$ is the feature embedding $f_e(x)$, although other representations may be used. Typically, a soft selection model $\hat{g} : \mathbb{R}^d \to [0, 1]$ is trained and $g$ is produced at inference time by choosing a threshold $\tau$ on $\hat{g}$ to achieve coverage level $\beta$ (i.e. $\mathbb{E}[\mathbf{1}\{\hat{g}(X) \geqslant \tau\}] = \beta$).

**Calibration**. The model $f$ is said to be top-label calibrated if $\mathbb{E}_{\mathcal{D}}[y^c|\hat{f}(x) = p] = p$ for all $p \in [0, 1]$ in the range of $\hat{f}(x)$. To measure deviation from this condition, we calculate expected calibration error (ECE):

$$\text{ECE}_q = \left( \mathbb{E}_{\hat{f}(x)} \left[ \left( \left| \mathbb{E}_{\mathcal{D}}[y^c|\hat{f}(x)] - \hat{f}(x) \right| \right)^q \right] \right)^{\frac{1}{q}} \tag{1}$$

where $q$ is typically 1 or 2. A recalibrator model $h$ can be applied to $f$ to produce outputs in the interval $[0, 1]$ such that $h(f(x)) \in \mathbb{R}^K$ is the recalibrated prediction confidence for input $x$ and $\hat{h}(f(x)) = \max_k h(f(x))_k$. See Section 4.3 for details on specific forms of $h(\cdot)$.

**Selective Calibration**. Under the notion of calibrated selective classification introduced in Fisch et al. (2022), a predictor is selectively calibrated if $\mathbb{E}_{\mathcal{D}}\left[y^c|\hat{f}(x) = p, g(x) = 1\right] = p$ for all $p \in [0, 1]$ in the range of $\hat{f}(x)$ where $g(x) = 1$. To interpret this statement, for the subset of examples that are accepted (i.e. $g(x) = 1$), for a given confidence level $p$ the predicted label should be correct for $p$ proportion of instances. Selective expected calibration error is then:

$$\text{S-ECE}_q = \left( \mathbb{E}_{\hat{f}(x)} \left[ \left( \left| \mathbb{E}_{\mathcal{D}}[y^c|\hat{f}(x), g(x) = 1] - \hat{f}(x) \right| \right)^q | g(x) = 1 \right] \right)^{\frac{1}{q}} \tag{2}$$

It should be noted that selective calibration is a separate goal from selective accuracy, and enforcing it may in some cases decrease accuracy. See Appendix C.2 for a more thorough discussion and empirical results regarding this potential tradeoff. Here we are only concerned with calibration, and leave methods for exploring the Pareto front of selective calibration and accuracy to future work.

## 4 SELECTIVE RECALIBRATION

In order to achieve lower calibration error, we propose jointly optimizing a selection and a recalibration model. Expected calibration error under both selection and recalibration is equal to

$$\text{SR-ECE}_q = \left( \mathbb{E}_{\hat{h}(f(x))} \left[ \left( \left| \mathbb{E}_{\mathcal{D}}[y^c|\hat{h}(f(x)), g(x) = 1] - \hat{h}(f(x)) \right| \right)^q | g(x) = 1 \right] \right)^{\frac{1}{q}}. \tag{3}$$

Taking SR-ECE$_q$ as our loss quantity of interest, our goal in selective recalibration is to solve the optimization problem:

$$\min_{g,h} \text{SR-ECE}_q \quad \text{s.t.} \quad \mathbb{E}_{\mathcal{D}}[g(x)] \geqslant \beta \tag{4}$$

where $\beta$ is our desired coverage level. Intuitively, to optimize the quantity in Eq. 4, one could apply only $h$ or $g$, first train $h$ and then $g$ (or vice versa), or jointly train $g$ and $h$ (i.e. selective recalibration). In Fisch et al. (2022), only $g$ is applied; however, as our experiments will show, much of the possible reduction in S-ECE comes from $h$. While $h$ can be effective alone, typical recalibrators are inexpressive, and thus may benefit from rejecting some difficult-to-fit portion of the data (as we find to be the case in experiments on several real-world datasets in Section 5). Training the models sequentially is also sub-optimal, as the benefits of selection with regards to recalibration can only be realized if the two models can interact in training, since fixing the first model constrains the available solutions.

Selective recalibration, where $g$ and $h$ are trained together, admits any solution available to these approaches, and can produce combinations of $g$ and $h$ that are unlikely to be found via sequential optimization (we formalize this intuition theoretically via an example in Section 6). Since Eq. 4 cannot be directly optimized, we instead follow Geifman & El-Yaniv (2019) and Fisch et al. (2022) and define a surrogate loss function $L$ including both a selective error quantity and a term to enforce the coverage constraint:

$$\min_{g,h} L = L_{sel} + \lambda L_{cov}(\beta) \tag{5}$$

We describe choices for $L_{sel}$ (selection loss) and $L_{cov}$ (coverage loss) in Sections 4.1 and 4.2, along with calibrator models in Section 4.3. Finally, we specify the inference procedure in Section 4.4, and explain how the soft constraint in Eq. 5 is used to satisfy Eq. 4.

### 4.1 SELECTION LOSS

In selective recalibration, the selection loss term measures the calibration of selected instances. Its general form for a batch with $n$ examples is:

$$L_{sel} = \frac{l(f, \hat{g}, h, x, y)}{\frac{1}{n} \sum_i \hat{g}(x_i)} \tag{6}$$

where $l$ measures the loss on selected examples and the denominator scales the loss according to the proportion preserved. We consider 3 forms of $l$: S-MMCE, S-MCE, and S-TLBCE.

**S-MMCE**, introduced in (Fisch et al., 2022), is a kernel-based loss which penalizes pairs of instances that have similar confidence and both are far from the true label (see Appendix A.1).

**S-MCE** is a selective version of a typical multi-class cross entropy loss:

$$l_{\text{S-MCE}}(f, \hat{g}, h, x, y) = \frac{-1}{n} \sum_i \hat{g}(x_i) \log h(f(x_i))_{y_i} \tag{7}$$

In the case that the model is incorrect ($y^c = 0$), this loss will penalize based on under-confidence in the ground truth class. However, our goal is calibration according to the *predicted* class. We thus propose a loss function for training a selective recalibration model given a black box predictor $f$ that we do not wish to alter, based on the typical approach to optimizing a smooth recalibration model,

**S-TLBCE (Selective Top-Label Binary Cross Entropy)**:

$$l_{\text{S-TLBCE}}(f, \hat{g}, h, x, y) = \frac{-1}{n} \sum_i \hat{g}(x_i) \Big[ y_i^c \log \hat{h}(f(x_i)) + (1 - y_i^c) \log(1 - \hat{h}(f(x_i))) \Big] \tag{8}$$

In contrast to S-MCE, in the case of an incorrect prediction S-TLBCE penalizes based on over-confidence in the predicted label. While in the binary case these losses are the same, in the multi-label case S-TLBCE aligns with our typical notion of top-label calibration error, as well as the typical Platt or temperature scaling objectives, and makes this a natural loss function for training a selective recalibration model.

## 4.2 COVERAGE LOSS

We employ a coverage loss that targets a specific $\beta$, $L_{cov}(\beta) = \left(\beta - \frac{1}{n}\sum_i \hat{g}(x_i)\right)^2$. Here we assume the user aims to reject that proportion of the data. Note that unlike selective accuracy, which aims to reject as little data as possible, selective calibration is not monotonic w.r.t. individual examples, so a mismatch in coverage between training and deployment may hurt test performance. Alternatively, Fisch et al. (2022) use a logarithmic regularization approach for enforcing the coverage constraint without a specific target $\beta$, computing cross entropy between the output of $\hat{g}$ and a target vector of all ones. However, we found this approach to be unstable and sensitive to the choice of $\lambda$ in initial experiments, while our coverage loss enabled stable training at any sufficiently large choice of $\lambda$, similar to the findings of Geifman & El-Yaniv (2019).

## 4.3 RECALIBRATION MODELS

We consider two differentiable and popular calibration models, Platt scaling and temperature scaling, both of which attempt to fit a function between model confidence and output correctness. The main difference between the models is that a Platt model works in the output probability space, whereas temperature scaling is applied to model logits before a softmax is taken. A Platt calibrator (Platt, 1999) produces output according to:

$$h^{\text{Platt}}(f(x)) = \frac{1}{1 + \exp(wf(x) + b)}$$

where $w, b$ are learnable scalar parameters. A temperature scaling model (Guo et al., 2017) produces output according to:

$$h^{\text{Temp}}(f(x)) = \text{softmax}\left(\frac{f_s(x)}{T}\right)$$

where $f_s(x)$ is the vector of logits (unnormalized scores) produced by $f$ and $T$ is the single learned (scalar) parameter. Both models are typically trained with a binary cross-entropy loss, where the labels 0 and 1 denote whether an instance is correctly classifier.

## 4.4 INFERENCE

Once we have trained $\hat{g}$ and $h$, we can flexibly account for $\beta$ by selecting a threshold $\tau$ on unlabeled test data (or some other representative tuning set) such that $\mathbb{E}[\mathbf{1}\{\hat{g}(X) \geqslant \tau\}] = \beta$. The model $g$ is then simply $g(x) := \mathbf{1}\{\hat{g}(x) \geqslant \tau\}$. A separate procedure for calibrating $\tau$ according to rigorous high-probability coverage guarantees is described in Appendix A.

## 5 EXPERIMENTS

Now we examine the performance of selective recalibration and baselines when applied to models pre-trained on real-world datasets and applied to a target distribution possibly shifted from the training distribution. First, in Section 5.1, we investigate whether, given a small validation set of labeled examples drawn i.i.d. from the target distribution, joint optimization consistently leads to a low empirical selective calibration error. Subsequently, in Section 5.2 we study multiple out-of-distribution prediction tasks and the ability of a single system to provide decreasing selective calibration error across a range of coverage levels when faced with a further distribution shift from validation data to test data. We also analyze the tradeoff between selective calibration error and accuracy in this setting; those results are deferred to Appendix C.2.

For selective recalibration models, the input to $g$ is the feature embedding. Temperature scaling is used for multi-class examples and Platt scaling is applied in the binary cases (following Guo et al. (2017) and initial results on validation data). Calibration error is measured using ECE-1 and ECE-2 with equal mass binning, which has been shown to have lower bias (Roelofs et al., 2020) than binning with equal width. For the selection loss, we use $l_{\text{S-TLBCE}}$ and $l_{\text{S-MMCE}}$ for binary tasks, and include a selective version of typical multi-class cross-entropy $l_{\text{S-MCE}}$ for multi-class tasks. One epoch of pre-training is performed where $g$ is fixed to output 1 in order to reasonably initialize the calibrator parameters before the selector is trained. Models are trained both with $h$ fixed after this pre-training epoch (denoted as "sequential" in results) and when it is jointly optimized throughout

training (denoted as "joint" in results). Our selection baselines include confidence-based rejection ("Confidence") and multiple out-of-distribution (OOD) detection methods ("Iso. Forest", "One-class SVM"). All selection baselines are applied to the recalibrated model in order to make the strongest comparison. We make further comparisons to recalibration baselines, including temperature scaling and Platt scaling, which have been described previously. We also include binning-based re-calibrators: histogram binning and Platt binning (Kumar et al., 2019). See Appendix B for more experiment details including calibration error measurement and baseline implementations.

## 5.1 SELECTIVE RECALIBRATION WITH I.I.D. DATA

First, we test whether selective recalibration consistently produces low S-ECE in a setting where there is a validation set of labeled training data available from the same distribution as test data. We use the Camelyon17 (Bandi et al., 2018) dataset with 1000 samples and a Platt scaling $h$, as well as the ImageNet dataset with 2000 samples and temperature scaling $h$, and report selective calibration error for coverage level $\beta \in \{0.75, 0.8, 0.85, 0.9\}$. Our soft selector $\hat{g}$ is a shallow fully-connected network (2 hidden layers with dimension 128). ECE-2 results are displayed in Figure 2, with ECE-1 results leading to similar conclusions, shown in Appendix C.1. Full experiment details, including model specifications and training parameters, can be found in Appendix B.

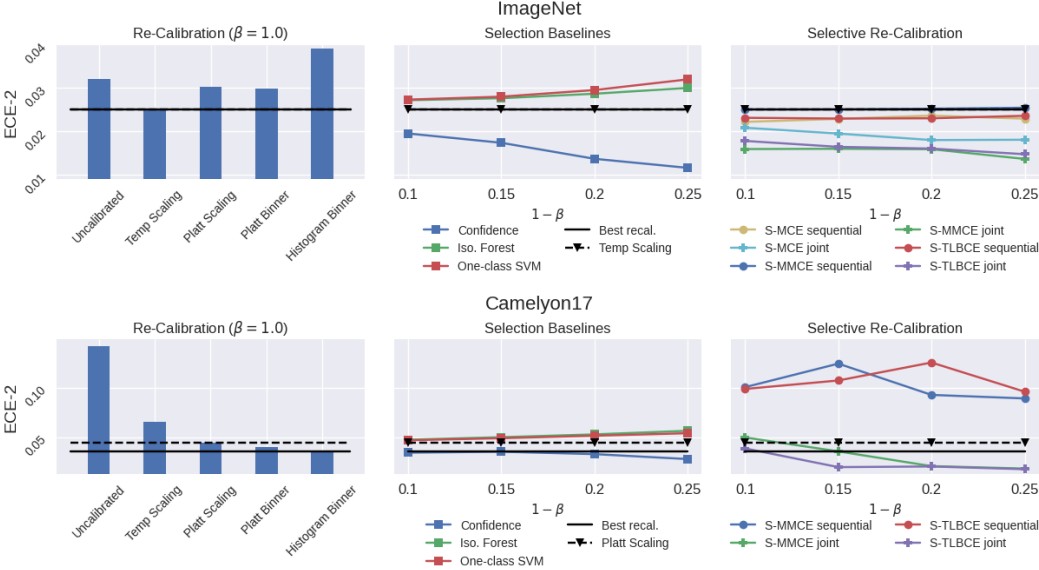

Figure 2: Selective ECE-2 on ImageNet and Camelyon17 for coverage level $\beta \in \{0.75, 0.8, 0.85, 0.9\}$. **Left**: Various recalibration methods are trained using labeled validation data; the top performer establishes a baseline. **Middle**: Selection baselines including confidence-based rejection and various OOD measures. **Right**: Selective recalibration with different loss functions.

Our results show that by jointly optimizing the selector and recalibration models, we are able to achieve consistently improved ECE at the given coverage level $\beta$ compared to first training $h$ and then $g$. In addition, the S-TLBCE loss shows more consistent performance than S-MMCE, as it reduces ECE-2 (and ECE-1, see Appendix Fig. 4) in every case, whereas training with S-MMCE increases ECE in some cases. We also find selective recalibration achieves the lowest ECE in almost every case in comparison to recalibration alone. While the confidence-based rejection strategy performs well in these experiments, this is not a good approach to selective calibration in general, as this is a heuristic strategy and may fail in cases where a model's confident predictions are in fact poorly calibrated (see Section 5.2 for examples).

## 5.2 SELECTIVE RE-CALIBRATION UNDER DISTRIBUTION SHIFT

In this experiment, we study the various methods applied to a multi-class medical imaging problem, RxRx1, as well as zero-shot image classification with CLIP and CIFAR-100-C. We follow the setting

of Fisch et al. (2022) where the test data is drawn from a shifted distribution with respect to the validation set and the goal is not to target a specific $\beta$, but rather to train a selector that works across a range of coverage levels. In the case of RxRx1 there is a strong batch processing effect which leads to a 9% difference in accuracy between validation (18%) and test (27%) data, and we also apply a selective recalibration model trained on validation samples from CIFAR-100 to test examples drawn from CIFAR-100-C. Our validation sets have 1000 (RxRx1) or 2000 (CIFAR-10) examples, $\hat{g}$ is a small network with 1 hidden layer of dimension 64, and we set $\beta = 0.5$ when training the models. For our results we report the area under the curve (AUC) for the coverage vs. error curve, a typical metric in selective classification (Geifman & El-Yaniv, 2017; Fisch et al., 2022) that reflects how a model can reduce the error on average at different levels of $\beta$. We measure AUC in the range $\beta = [0.5, 1.0]$, with measurements taken at intervals of 0.05 (i.e. $\beta \in [0.5, 0.55, 0.6, ..., 0.95, 1.0]$). Additionally, to induce robustness to the distribution shift we noise the selector/recalibrator input. See Appendix B for full specifications.

Results are shown in Table 1. First, these results highlight that even in this OOD setting, the selection-only approach of Fisch et al. (2022) is not enough and recalibration is a key ingredient in improving selective calibration error. Fixing $h$ and then training $g$ performs better than joint optimization for RxRx1, likely because the distribution shift significantly changed the optimal temperature for the region of the feature space where $g(x) = 1$. Joint optimization performs best for CIFAR-100-C, and does still significantly improve S-ECE on RxRx1, although it's outperformed by fixing $h$ first in that case. The confidence baseline performs quite poorly on both experiments and according to both metrics, significantly increasing selective calibration error in all cases.

| Selection | Opt. of $h, g$ | RxRx1 | | CIFAR-100-C | |
|---|---|---|---|---|---|
| | | ECE-1 | ECE-2 | ECE-1 | ECE-2 |
| Confidence | - | 0.071 | 0.081 | 0.048 | 0.054 |
| OSVM | - | 0.058 | 0.077 | 0.044 | 0.051 |
| Iso. Forest | - | 0.048 | 0.061 | 0.044 | 0.051 |
| S-MCE | sequential | 0.059 | 0.075 | 0.033 | 0.041 |
| | joint | 0.057 | 0.073 | 0.060 | 0.068 |
| S-MMCE | sequential | **0.036** | **0.045** | 0.030 | 0.037 |
| | joint | **0.036** | **0.045** | 0.043 | 0.051 |
| S-TLBCE | sequential | **0.036** | **0.045** | 0.030 | 0.037 |
| | joint | 0.039 | 0.049 | **0.026** | **0.032** |
| Recalibration | Temp. Scale | 0.055 | 0.070 | 0.041 | 0.047 |
| ($\beta = 1.0$) | None | 0.308 | 0.331 | 0.071 | 0.079 |

Table 1: RxRx1 and CIFAR-100-C AUC in the range $\beta = [0.5, 1.0]$.

## 6 THEORETICAL ANALYSIS

To build a deeper understanding of selective recalibration (and its alternatives), we consider a theoretical situation where a pre-trained model is applied to a target distribution different from the distribution on which it was trained, mirroring both our experimental setting and a common challenge in real-world deployments. We show that with either selection or recalibration alone there will still be calibration error, while selective recalibration can achieve ECE = 0. We also show in Appendix D that joint optimization of $g$ and $h$, as opposed to sequentially optimizing each model, is necessary to achieve zero calibration error.

### 6.1 SETUP

We consider a setting with two classes, and without loss of generality we set $y \in \{-1, 1\}$. We are given a classifier pre-trained on a mixture model, a typical way to view the distribution of objects in images (Zhu et al., 2014; Thulasidasan et al., 2020). The pre-trained classifier is then applied to a target distribution containing a portion of outliers from each class unseen during training. Our specific choices of classifier and training and target distributions are chosen for ease of interpretation and analysis; however, the intuitions built can be applied more generally.

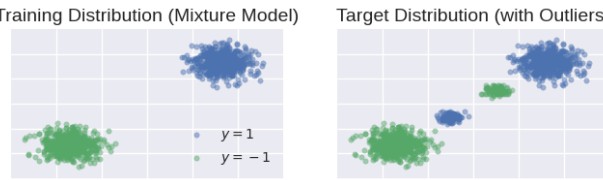

Figure 3: A classifier pre-trained on a mixture model is applied to a target distribution with outliers.

### 6.1.1 DATA GENERATION MODEL

**Definition 1** (Target Distribution). *The target distribution is defined as a $(\theta^*, \sigma, \alpha)$-perturbed mixture model over $(x, y) \in \mathbb{R}^p \times \{1, -1\}$: $x \mid y \sim zJ_1 + (1 - z)J_2$, where $y$ follows the Bernoulli distribution $\mathbb{P}(y = 1) = \mathbb{P}(y = -1) = 1/2$ and $z$ follows a Bernoulli distribution $\mathbb{P}(z = 1) = \beta$.*

Our data model considers a mixture of two distributions with disjoint and bounded supports, $J_1$ and $J_2$, where $J_2$ is considered to be an outlier distribution. Specifically, for $y \in \{-1, 1\}$, $J_1$ is supported in the balls with centers $y\theta^*$ and radius $r_1$, $J_2$ is supported in the balls with centers $-y\alpha\theta^*$ and radius $r_2$, and both $J_1$ and $J_2$ have standard deviation $\sigma$. See Figure 3 for an illustration of our data models, and Appendix D.1 for a full definition of the distribution.

### 6.1.2 CLASSIFIER ALGORITHM

Recall that in our setting $f$ is a pre-trained model, where the training distribution is unknown and we only have samples from some different target distribution. We follow this setting in our theory by considering a (possibly biased) estimator $\hat{\theta}$ of $\theta^*$, which is the output of a training algorithm $\mathscr{A}(S^{tr})$ that takes the i.i.d. training data set $S^{tr} = \{(x_i^{tr}, y_i^{tr})\}_{i=1}^m$ as input. The distribution from which $S^{tr}$ is drawn is **different** from the target distribution from which we have data to train the selection and recalibration models. We only impose one assumption on the model $\hat{\theta}$: that $\mathscr{A}$ outputs a $\hat{\theta}$ that will converge to $\theta_0$ if training data is abundant enough and $\theta_0$ should be aligned with $\theta^*$ with respect to direction (see Assumption 3, Appendix D.3 for formal statement). For ease of analysis and explanation, we consider a simple classifier defined by $\hat{\theta} = \sum_{i=1}^m x_i^{tr} y_i^{tr}/m$ when the training distribution is set to be an unperturbed Gaussian mixture $x^{tr}|y^{tr} \sim \mathcal{N}(y^{tr} \cdot \theta^*, \sigma^2 I)$ and $y^{tr}$ follows a Bernoulli distribution $\mathbb{P}(y^{tr} = 1) = 1/2$.[1] This form of $\hat{\theta}$ is closely related to Fisher's rule in linear discriminant analysis for Gaussian mixtures (see Appendix D.3.1 for further discussion).

Having obtained $\hat{\theta}$, our classifier aligns with the typical notion of a softmax response in neural networks. We first obtain the confidence vector $f(x) = (f_1(x), f_{-1}(x))^\top$, where

$$f_{-1}(x) = \frac{1}{e^{2\hat{\theta}^\top x} + 1}, \quad f_1(x) = \frac{e^{2\hat{\theta}^\top x}}{e^{2\hat{\theta}^\top x} + 1}. \tag{9}$$

and then output $\hat{y} = \operatorname{argmax}_{k \in \{-1, 1\}} f_k(x)$. For $k \in \{-1, 1\}$, the confidence score $f_k(x)$ represents an estimator of $\mathbb{P}(y = k|x)$ and the final classifier is equivalent to $\hat{y} = \operatorname{sgn}(\hat{\theta}^\top x)$.

### 6.2 MAIN THEORETICAL RESULTS

Having established our data and classification models, we now analyze why selective recalibration (i.e., joint training of $g$ and $h$) can outperform recalibration and selection performed alone or sequentially. For recalibration, we focus on studying the popular temperature scaling model[2] and we consider $\text{ECE}_q$ with $q = 1$ (and drop the subscript $q$ for notational simplicity below).

We study the following ECE quantities for recalibration alone, selection alone, and selective recalibration calculated according to our data model.[3] For the clarity of theorem statements and proofs,

---

[1] The in-distribution case also works under our data generation model.

[2] For Platt scaling, the analysis is basically the same.

[3] By studying population quantities, our analysis is not dependent on any binning-methods that are commonly used in empirically calculating expected calibration errors.

we will restate definitions such as recalibration error and selective calibration error under temperature rescaling to make them explicitly dependent on $g$ and parameter $T$ and tailored for the binary case we are studying. We want to emphasize that we are ***not introducing new concepts***, but instead offering different surface forms of the same quantities.

- The expected calibration error after incorporating temperature re-scaling with parameter $T$ is defined as: R-ECE$(T) := \mathbb{E}_{\hat{f}(x)} |\mathbb{P}[y = \hat{y} \mid \hat{f}(x) = pT] - p|$.

- The expected calibration error after incorporating a selection model $g$ is defined as: S-ECE$(g) := \mathbb{E}_{\hat{f}(x)} \left[ |\mathbb{P}[y = \hat{y} \mid \hat{f}(x) = p, g(x) = 1] - p| \big| g(x) = 1 \right]$.

- With a slight abuse of notation, we denote the expected calibration error after applying a selective recalibration model including the selection model $g$ and temperature parameter $T$ as: SR-ECE$(g, T) := \mathbb{E}_{\hat{f}(x)} \left[ |\mathbb{P}[y = \hat{y} \mid \hat{f}(x) = pT, g(x) = 1] - p| \big| g(x) = 1 \right]$.

Our first theorem proves that under our data generation model, S-ECE and R-ECE can never reach $0$, but SR-ECE can reach $0$ by choosing appropriate $g$ and $T$.

**Theorem 1.** *Under Assumption 3, for any $\delta \in (0, 1)$ and $\hat{\theta}$ output by $\mathscr{A}$, there exist thresholds $M \in \mathbb{N}^+$ and $\tau > 0$ such that if $\max\{r_1, r_2, \sigma, \|\theta^*\|\} < \tau$ and $m > M$, there exists a positive lower bound $L$, with probability at least $1 - \delta$ over $S^{tr}$:*

$$\min \big\{ \min_{g: \mathbb{E}[g(x)] \geq \beta} \text{S-ECE}(g), \quad \min_{T \in \mathbb{R}} \text{R-ECE}(T) \big\} > L.$$

*However, there exists $g_0$ satisfying $\mathbb{E}[g_0(x)] \geq \beta$ and $T_0$, such that SR-ECE$(g_0, T_0) = 0$.*

**Intuition and interpretation.** Here we give some intuition for understanding our results. Under our data generation model, R-ECE is expressed as:

$$\text{R-ECE}(T) = \mathbb{E}_{v = \hat{\theta}^\top x} \left| \frac{\mathbf{1}\{v \in \mathcal{A}\}}{1 + \exp\left(\frac{-2\alpha \hat{\theta}^\top \theta^*}{\sigma^2 \|\hat{\theta}\|^2} \cdot v\right)} + \frac{\mathbf{1}\{v \in \mathcal{B}\}}{1 + \exp\left(\frac{2\alpha \hat{\theta}^\top \theta^*}{\sigma^2 \|\hat{\theta}\|^2} \cdot v\right)} - \frac{1}{e^{-2v/T} + 1} \right|$$

for disjoint sets $\mathcal{A}$ and $\mathcal{B}$ which correspond to points on the support of $J_1$ and $J_2$ respectively. In order to achieve zero R-ECE, when $v \in \mathcal{A}$, we need $T = \hat{\theta}^\top \theta^* / (\sigma^2 \|\hat{\theta}\|^2)$. However, for $v \in \mathcal{B}$ we need $T = -\alpha \hat{\theta}^\top \theta^* / (\sigma^2 \|\hat{\theta}\|^2)$. These clearly cannot be achieved simultaneously. Thus the presence of the outlier data makes it impossible for the recalibration model to properly calibrate the confidence for the whole population. A similar expression can be obtained for S-ECE. As long as $\hat{\theta}^\top \theta^* / (\sigma^2 \|\hat{\theta}\|^2)$ and $-\alpha \hat{\theta}^\top \theta^* / (\sigma^2 \|\hat{\theta}\|^2)$ are far from $1$ (i.e., miscalibration exists), no choice of $g$ can reach $0$ S-ECE. In other words, no selection rule alone can lead to calibrated predictions, since no subset of the data is calibrated under the pre-trained classifier. However, by setting $g_0(x) = 0$ for all $x \in \mathcal{B}$ and $g_0(x) = 1$ otherwise, and choosing $T_0 = \hat{\theta}^\top \theta^* / (\sigma^2 \|\hat{\theta}\|^2)$, SR-ECE is $0$. We conclude that achieving ECE $= 0$ on eligible predictions is only possible under selective recalibration, while selection or recalibration alone induce positive ECE. See Appendix D for more details and analysis. In Appendix D we also demonstrate that jointly learning a selection model $g$ and temperature scaling parameter $T$ can outperform sequential learning of $g$ and $T$ (see Theorem 2 for details).

## 7 CONCLUSION

We have shown both empirically and theoretically that combining selection and recalibration is a potent strategy for producing a set of well-calibrated predictions. Eight pairs of distribution and $\beta$ were tested when i.i.d. validation data is available. Selective recalibration with our proposed S-TLBCE loss function outperforms every single recalibrator tested in 7 cases, and always reduces S-ECE with respect to the calibrator employed by the selective recalibration model itself. Taken together, these results show that while many popular recalibration functions are quite effective at reducing calibration error, they can often be better fit to the data when given the opportunity to ignore a small portion of difficult examples. Thus, in domains where calibrated confidence is critical to decision making, selective recalibration is a practical and lightweight strategy for improving outcomes downstream of deep learning model predictions.

ETHICS STATEMENT

While the goal of our method is to foster better outcomes in settings of societal importance like medical diagnosis, selective classification has been shown to increase disparities in accuracy across protected groups (Jones et al., 2020). Work has been done to mitigate this effect in both classification (Jones et al., 2020) and regression (Shah et al., 2021) tasks, for example by enforcing calibration across groups. Future work on selective recalibration could focus on analyzing and mitigating any unequal effects of the algorithm.

REPRODUCIBILITY

We intend to release all code used in our experiments upon the publication of this paper. All pretrained models and datasets are publicly available, and all experiment parameters appear in the code as well as either Section 5 or Appendix B.

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
