# OpenReview forum: "Making Predictors More Reliable with Selective Recalibration"
_ICLR.cc/2024/Conference — Submitted to ICLR 2024_

### Official Review · Reviewer_RXXm · 2023-10-26

**Soundness:** 3 good
**Presentation:** 3 good
**Contribution:** 2 fair
**Rating:** 5
**Confidence:** 4

**Summary:**

This paper proposes an extension to a recent line of work on calibrated selective classification (Fisch et al., 2022), which consists of selectively predicting inputs so as to minimize calibration error. The authors' method jointly optimizes a selection function along with a recalibration model, so as to minimize calibration error on the selected subset of the data. They show that this approach can significantly outperform doing only selective prediction, and also show that it can outperform selection and recalibration done in sequence.

**Strengths:**

- **Originality:** The paper introduces an objective that combines selective classification and recalibration (S-TLBCE, Equation 8) and proves for a simple class of data distributions that the joint procedure can outperform each individual procedure. While the objective and analysis seem new, the practical advancement of this work is a straightforward extension of [1].
- **Quality:** Overall the quality of the work is good, with the problem of selective calibration being well-described/well-motivated and experiments designed appropriately to evaluate the proposed method.
- **Clarity:** The work is easy to follow, with the experiments and theory described with adequate detail for the most part.
- **Significance:** Selective classification optimizing for calibration is relatively new, being (to the best of my knowledge) largely introduced in the recent (2022) work of [1]. This paper constitutes a natural extension of [1], and while the empirical results correspond to marked improvements in some regimes, I have some reservations regarding the overall contribution of the work in the context of [1].

[1] https://arxiv.org/abs/2208.12084

**Weaknesses:**

## Weaknesses
1. **Novelty/improvement of proposed approach.** The approach is conceptually a minor change to the methodology of [1]. While S-TLBCE seems like a new objective in this context, it does not perform better (in fact it even performs worse on ImageNet) than the jointly optimized version of S-MMCE from [1], except for the OOD tests on CIFAR-100-C. Furthermore, the table describing these latter experiments (Table 1) is unclear - the description claims to be reporting AUC over various coverage levels but only ECE-1 and ECE-2 are reported? Also, the naive confidence-based rejection strategy (which should be described in the main paper) performs very well on the ImageNet/Camelyon17 experiments - the authors say that this strategy falls apart in the OOD case, but is this considering confidence-based rejection with recalibration (in sequence)?

## Recommendation
Overall I think the paper is tackling an interesting and relatively new problem, but I feel the contribution is too marginal in its current form to warrant a clear acceptance. Furthermore, the experimental results do not seem to me to be convincingly better than the previous work of [1], and for this reason I recommend **weak reject**.

[1] https://arxiv.org/abs/2208.12084

**Questions:**

- Several questions are stated in weaknesses above.
- Figure 1 is difficult to understand. What subsets of the data do the blue and green curves correspond to? Shouldn't there only be one curve for 1(a) and 1(b) since no selection is being done for these?
- The definition of R-ECE seems a bit strange to me; linearly rescaling the predicted *probabilities* by $T$ will not be the same as rescaling the predicted scores, so this is not exactly temperature scaling.
- The ECE calculations in the appendix are hard to follow due to the presentation, improving spacing/detail would help here.

---

> ### Author Response · Authors · 2023-11-15
> **Author Rebuttal**
>
> Thank you for taking the time to read and offer feedback on our work.  Please see our response to your concerns below.
>
>  - Please see the response to all reviewers for an explanation of the novelty/performance improvement of our method as compared to [1].
>  - S-TLBCE shows more consistent performance than S-MMCE.  See Table 1 CIFAR-100-C results for an example of S-MMCE failing, whereas S-TLBCE always improves on the recalibration baseline.
>  - We are also going to include Brier Score results for all experiments in the next version of this paper.
>  - Recalibration was applied in tandem with the confidence-based rejection strategy.
>
> With respect to the contribution, once again we point to our comment to all reviewers above.  If the concern is performance in particular, then we believe that our algorithm is quite significant, since we show that considering recalibration along with selection is a necessary step in achieving the best selective calibration error, while the importance of recalibration is mostly ignored by [1].

---

> > ### Comment · Reviewer_RXXm · 2023-11-17
> > **Response**
> >
> > Thank you for taking the time to respond to my comments.
> > - I understand that the proposal of joint vs sequential makes a significant difference, and this is definitely valuable. My concern with novelty was more-so that, conditioned on joint optimization, the value of S-TLBCE over S-MMCE seems marginal. One way to improve this could be more exploration of the OOD setting where the gap seems to be more significant between the two, but I understand there is not a lot of time left to run those experiments during the review period.
> > - My questions about Figure 1, as well as the definition of R-ECE remain.

---

> > > ### Author Response · Authors · 2023-11-20
> > > **Author Response**
> > >
> > > Thank you for your reply and the suggestion for strengthening the empirical investigation of S-TLBCE. We apologize that we missed the R-ECE and Figure 1 questions earlier.  Please see answers below.
> > >
> > >  - Figure 1: The blue and green curves represent two different subgroups that exist in the full data distribution.  We will add further explanation for this figure.
> > >  - R-ECE: Sorry for the confusion, we meant to say the score is $p^T$ and the the prediction is $f(p^T)$, as we wrote in the appendix. Sorry about the typo.

---

### Official Review · Reviewer_6mVa · 2023-10-31

**Soundness:** 2 fair
**Presentation:** 2 fair
**Contribution:** 2 fair
**Rating:** 5
**Confidence:** 2

**Summary:**

This paper proposes to combine selective prediction and post-hoc calibration to achieve more reliable performance for classification tasks. The proposed optimization framework is based on another recent work which combines selective prediction and training-time calibration. In the proposed selective recalibration framework, a new loss function, Selective Top-Label Binary Cross Entropy (S-TLBCE) is proposed for training such a recalibrator and a selective model at the same time. The proposed approach is evaluated on real-world medical diagnosis datasets and image classification datasets, and the results show its effectiveness.

**Strengths:**

1. The idea of combining selective prediction and model calibration is very useful and realistic in many safety aware tasks. Also, previous work studied the combination of selective prediction and train-time calibration, the combination of selective prediction and recalibration seems more important as post-hoc calibration achieves better calibration performance generally.

2. Except for the experiments on real-world datasets, the authors also give theoretic results based on a simple and intuitive data generation model.

**Weaknesses:**

1. The technical novelty and overall contribution is quite limited, primarily because the combination of selective prediction and model calibration has been studied in previous work and the proposed method is a straightforward combination of existing optimization framework and recalibration model.

2. The writing and organization of this paper are not good enough. Many places involving notations are confusing, for example, some loss functions appear in the Methodology section but are not used in the following optimization framework, and some notations are not defined before their first appearance.

3. The empirical evaluation is also limited, as a result, the effectiveness and soundness cannot be sufficiently shown. For example, some benchmark datasets for image classification, which are commonly used in the context of model calibration like SVHN/CIFAR-10, are not used in experiment section. Moreover, the experimental part does not provide sufficient information about the dataset used and the way it was divided (for training and validation), as well as the model structure.

**Questions:**

Please refer to the weaknesses section.

---

> ### Author Response · Authors · 2023-11-15
> **Author Rebuttal**
>
> We appreciate your feedback.  Here we respond to the specific concerns that you have raised.
>
>  - Please see the note to all reviewers for a detailed explanation of the differences with previous work.  We believe that the connection between selection and recalibration is important and unexplored, and that our algorithm results in major performance improvements over the approach taken in Fisch et al.
>  - Could you please point to specific examples where the notation is confusing?  Also, which loss functions are introduced that are not used?  We appreciate the feedback on the presentation, but would please need more specific information in order to use it to improve the paper.
>  - While we appreciate this suggestion with respect to evaluation, we strongly disagree here.  We produced results for 10-15 models (including baselines) across 2 important settings with 2 datasets each.  While we could include an MLP recalibrator and significance testing as suggested by the other reviewers, we do not believe that the current evaluation is limited.  Additionally, we think that the datasets that we explored are much better than SVHN/CIFAR-10.  These are typically treated as toy datasets at this point in the evolution of deep learning, and are much less challenging than either image classification dataset that we used (CIFAR-100, ImageNet).  We also believe that our inclusion of the medical tasks (Camelyon, RxRx1) is much more important, as we contend that this method is appropriate for the medical domain.  Since we are limited in space, switching these out for SVHN/CIFAR-10 would significantly weaken our results.  The validation and test sets are randomly drawn, we will be more explicit about this in an updated draft.  The model structure is described in the appendix, however we will consider whether more detail can be added there (or moved to the main paper).

---

> > ### Comment · Reviewer_6mVa · 2023-11-22
> > **Thanks for the response**
> >
> > Thanks for the response.
> >
> > - After reading the general response regarding the novelty and the distinctions from previous work, my concerns about the novelty and contribution remain. Given the work in [On Calibration of Modern Neural Networks, https://arxiv.org/abs/1706.04599], which suggest that recalibration can improve calibration performance with very large margin, applying recalibration on the Calibrated Selective Classification framework is too straightforward.
> >
> >
> > - There are some notations like $\lambda$ in Eq.(5), {$\theta^*, \epsilon, \alpha$} in Definition 1 (also, indicating that their range would be better) are not defined before their first appearance. Eq. (5-8) introduce the overall loss function in a hierarchical structure. However, the representation of each term in the loss function is quite complex, which may lead readers to misunderstand whether a specific term is used in the optimization objective. I suggest that instead of dedicating separate sections for each term in the loss function, the authors can consider presenting them in a more integrated manner. For instance, coverage loss may not require its own section but could be introduced directly below Eq.(5).  Consequently, Eq.(5) could be revised as follows:
> >
> > $$\qquad\qquad\min _{g, h} L _\phi = \frac{\phi(f, \hat{g}, h, x, y)}{\frac{1}{n} \sum_i \hat{g}\left(x_i\right)} + \left(\beta-\frac{1}{n} \sum_i \hat{g}\left(x_i\right)\right)^2,$$
> >
> > &nbsp;&nbsp;&nbsp;&nbsp;&nbsp;&nbsp;&nbsp;where the specific choice of $\phi$ can be indicated in the the overall loss term $ L _\phi$.
> >
> > - Furthermore, in $h^{\text {Platt }}(f(x))=\frac{1}{1+\exp (w f(x)+b)}$, it seems that $w$ and $b$ should be a vector. Further clarification for this or distinguishing vectors from scalars by bolding can enhance the clarity of symbols.
> >
> > - I agree that the used datasets are more challenging, and the authors‘ argument about the use of additional space with other datasets weakening the results has convinced me.
> >
> > Based on the author's response, I raise my rating to 5.

---

> > > ### Author Response · Authors · 2023-11-22
> > > **Thank you**
> > >
> > > Thank you for the response and raising your score.  We appreciate the feedback on our presentation, and will use it to update our paper.

---

### Official Review · Reviewer_DcsN · 2023-10-31

**Soundness:** 3 good
**Presentation:** 4 excellent
**Contribution:** 3 good
**Rating:** 8
**Confidence:** 5

**Summary:**

This paper extends the selective calibration setting of Fisch et. al. (2022) to allow for joint training of both selection _and_ recalibration functions. Reliable confidence estimation is a key property for predictors operating in sensitive domains, but is not typically easy to achieve. Selective calibration combines "selective prediction" (i.e., allowing for abstention) with "calibration", in the sense that selectively calibrated predictors prioritize having calibrated predictions on the non-rejected population of inputs. Previous work by Fisch et. al. (2022) only considered optimizing the "selection" part given a fixed model. This paper follows the natural motivation of also training a recalibrator jointly, which is similar to previous work in standard selective classification such as SelectiveNet (Geifman and El-Yaniv, 2019). It is intuitive to see why such an approach would be a good idea for some input distributions and models (e.g., where the optimal calibrator across the full input space is not in the sigmoid family for Platt scaling, but is after selection). The authors also provide empirical and additional theoretical analysis by example that supports their joint design.

[1] Calibrated Selective Classification. Adam Fisch, Tommi Jaakkola, Regina Barzilay. 2022.

[2] SelectiveNet: A Deep Neural Network with an Integrated Reject Option. Yonatan Geifman, Ran El-Yaniv. 2019.

**Strengths:**

The paper is well written and its proposed method is well-motivated. Though it is a fairly straightforward extension of prior work (Fisch et. al. + Geifman and El-Yaniv), it also presents additional contribution in the form of a simplified selective calibration loss (S-TLBCE) which appears to work better in practice. The empirical and theoretical arguments for justifying joint vs. sequential training of $(g, h)$ is also useful.

**Weaknesses:**

I think that the paper does a good job at addressing the narrow question of improving selective calibration through joint selection _and_ recalibration. More broadly, I would have been excited to see a more nuanced approach to selective calibration (which Fisch et. al. also misses), in the sense that not all types of calibration error are necessarily equal. For example, under a selective recalibration framework, all of the following example rejections would be preferred (assuming that they don't have any other structure that makes recalibration easier post selection):

- reject(Predictions with confidence 95% but accuracy 90%) > reject(predictions with confidence 4% but accuracy 0%)
- reject(Predictions with confidence 80% but accuracy 100%) > reject(predictions with confidence 80% but accuracy 61%)

Obviously it depends on the application, but generally speaking it seems that rejecting the top-performing half of predictions simply because their confidences are slightly too high is not a useful strategy. Similarly for under- vs. over-confidence. Considering other calibration objectives (e.g., Decision Calibration from Zhao et. al., 2021) that are more expressive than ECE-1/ECE-2 could add significantly to this work's potential impact.

I'm also a bit worried about the data requirement for selective recalibration to work. If labeled examples from the target domain are not available (e.g., following the setting of Fisch et. al.), can the method still produce reliable rejections? Does recalibration on the training domain help here?

Minor formatting points:
- It would be helpful to keep equation numbering for all display equations.
- Unnumbered equation for $h^\mathrm{Temp}$ would benefit from \left \right parentheses.

[3] Calibrating Predictions to Decisions: A Novel Approach to Multi-Class Calibration. Shengjia Zhao, Michael P. Kim, Roshni Sahoo, Tengyu Ma, Stefano Ermon. 2021.

**Questions:**

- Figure 1 is a bit confusing to me. It's not easy to see what strategy the selective and selective recalibration algorithms take. The caption could use some additional clarification. For example, what is the desired coverage level? It seems like the basic strategy that is followed is to reject either the blue or the green (selective vs. selective recalibration, respectively). Though I would still expect a selective only approach to reject the mid-confidence blue examples disproportionately more than the upper/lower ranges (since this is where calibration error is highest per the reliability diagram).

- It makes sense that jointly learning selection and recalibration can help when the calibration error is too complex to be fit by the family of recalibration functions specified by Platt or temperature scaling. I'd be curious to see how this would compare to simply fitting a slightly more expressive family of calibrators (e.g., 2-layer NN), especially for data without distribution shifts.

- This is more of a half-baked suggestion than a question, but I'm curious if the authors ever experimented with the following setup. Suppose we modeled recalibration as a hard mixture of experts, where $h(x) = \sum_{i = 1}^{n} g_i(x) h_i(x)$ with $g(x)$ being 1-hot. On new distributions $p(x')$, one could "turn off" the lowest performing $g_i$ via rejection until $\sum_{i \in \text{rejected}} \mathbb{E}(g_i(x')) = \beta$. This is a generalization of the current setup with $g(x)$ being binary, and only training one $h(x)$ (as the other examples are discarded). On new distributions, however, dynamically reconfiguring this mixture (which will have been trained to fit subdomains in the training data) could prove to be effective, without the need for much new data to jointly recalibrate $h$.

**Details Of Ethics Concerns:**

None.

---

> ### Author Response · Authors · 2023-11-15
> **Author Rebuttal**
>
> Thank you for your positive feedback.  Below we respond to the particular questions that you raised.
>
>  - While it is only meant to be illustrative/high-level, we agree that Figure 1 is still a bit unclear.  The idea is that the desired coverage level is equal to the proportion of the data represented by the blue group.  Your suggestion regarding selection and the mid-confidence blue examples vs. upper/lower ranges is extremely helpful, and we will make this change.
>  - We agree that the expressive family of calibrators is a missing comparison, as we may be able to use the extra parameters for recalibration instead of selection.  We plan to implement and include this baseline in a future version of the work.
>  - Thank you for this suggestion regarding a MoE approach.  We have not explored this, but it sounds like it could be a very promising direction and we plan to explore it further in the future.

---

> > ### Comment · Reviewer_DcsN · 2023-11-22
> >
> > Thanks for the reply to my comments and questions. Just checking: were you planning to upload a new revision with those changes?

---

> > > ### Author Response · Authors · 2023-11-22
> > >
> > > We do not have a revised draft yet, but we plan to include the reviewers' useful feedback in the next version of the paper.

---

### Official Review · Reviewer_NrpW · 2023-11-01

**Soundness:** 3 good
**Presentation:** 3 good
**Contribution:** 2 fair
**Rating:** 3
**Confidence:** 3

**Summary:**

The submission proposes selective recalibration, which combines ideas from selective classification with calibration. The proposed method builds on work by Fisch et al. (2022) by optimizing rejection and calibration jointly. Experiments show that this yields better calibration for a given level of coverage than standard out-of-distribution detection methods applied separately to standard calibration methods. It also performs better in most cases than the "sequential" alternative to the joint optimization approach. Performance is evaluated for both i.i.d. settings and settings where an existing pretrained model is used in a new domain, without fine-tuning this model, by performing selective recalibration. There are also some theoretical results for a particular synthetic domain showing that better performance can be achieved with joint optimization.

**Strengths:**

The proposed approach makes sense, and it is plausible that joint optimization performs better than the alternatives considered.

**Weaknesses:**

One important baseline that is missing in the experiments is a calibration approach that is as complex as the multi-layer perceptron used in the proposed approach to perform selection. Platt scaling and temperature scaling, the calibration methods used in the experiments, have very few parameters, and discretization-based approaches are also very simple. The argument in the submission is that selective calibration can be beneficial because it may not be possible to achieve good fit for the calibration model to the entire data. However, an obvious approach to tackle this problem is to simply make the calibration model more complex: rather than using a linear logistic regression model as in Platt scaling, one can use a multi-layer perceptron instead.

I am also wondering about the sequential baseline used. A single epoch of calibration is performed before the selection model is trained. It is unclear why a single epoch is used. Also, it seems that improved performance could trivially be obtained by recalibrating again after the selection model has been trained. (This could be iterated, but that would probably be very similar to joint optimisation using gradient descent.)

No significance testing is performed and no confidence intervals are provided for estimated performance measures.

The first result in Section C.2 is disturbing: classification accuracy goes down with decreased coverage when applying the proposed method. It is unclear to me whether one would ever accept this in practical applications.

Other comments and typos:

"whether an instance is correctly classifier"

"outputs, We"

Section B.2.1: what is $\tilde f$?

Section B.4.2 does not mention validation data.

**Questions:**

N/A

---

> ### Author Response · Authors · 2023-11-15
> **Author Rebuttal**
>
> Thank you for your careful feedback.  Below we respond to the individual concerns that you have raised.
>
> - We appreciate the suggestion with respect to the MLP recalibration baseline, to see if the performance gain comes from simply having more parameters.  We plan to implement this and include the comparison in a final paper.
> - With respect to the single epoch of pretraining, this was a poor explanation on our part, which we plan to correct.  The Platt/temperature scaling models are optimized in the typical manner on the full validation set prior to the training of g.
> - We agree that significance testing should be included.  We will run more trials with random validation and test splits in order to better characterize the performance differences.
> - While we agree that it would be ideal if accuracy and calibration could improve together, we believe that in many important cases they are separate concerns, and classification accuracy is often not the primary concern.  For example, in other research we are working to develop a medical monitoring system to be deployed in ICU’s that predicts the probability of a serious condition developing.  This system will not output 0/1 predictions, but only calibrated risk scores.  We believe that many such human-in-the-loop decision scenarios exist, and that our algorithm can be extremely useful in those settings.  We will however add further exposition of this dynamic in a future version of the paper.

---

> > ### Comment · Reviewer_NrpW · 2023-11-23
> > **Response to authors' comments**
> >
> > Thank you very much for your response. However, to change my recommendation, at the very least, I would need to see the results of the suggested baseline as evidence that the increased number of parameters is not the reason for the observed improvements in performance.

---

> > > ### Author Response · Authors · 2023-11-23
> > > **Number of Recalibration Parameters**
> > >
> > > Thank you for the response.  We do note that the idea of recalibration with an increased number of parameters has been explored previously in the literature.
> > >
> > > In Guo 2017 [1], temperature scaling was compared to vector scaling and matrix scaling, each with an increased number of parameters.  They find that: "temperature scaling outperforms the vector and matrix Platt scaling variants, which are strictly more general methods. In fact, vector scaling recovers essentially the same solution as temperature scaling – the learned vector has nearly constant entries, and therefore is no different than a scalar transformation. In other words, network miscalibration is intrinsically low dimensional."
> > >
> > > In particular, their matrix scaling method in some cases leads to double the calibration error to start, and even fails to converge in the case of imagenet: "Matrix scaling performs poorly on datasets with hundreds of classes (i.e. Birds, Cars, and CIFAR-100), and fails to converge on the 1000-class ImageNet dataset. This is expected, since the number of parameters scales quadrat-
> > > ically with the number of classes. Any calibration model with tens of thousands (or more) parameters will overfit to a small validation set, even when applying regularization."
> > >
> > > While we agree this would be a useful baseline to include in our paper, the recalibration literature has been focused on these low dimensional solutions because they have been shown to work better, and we believe this point is largely taken for granted at this point.
> > >
> > > [1] https://arxiv.org/abs/1706.04599

---

> > > > ### Comment · Reviewer_NrpW · 2023-11-23
> > > > **New response**
> > > >
> > > > Matrix scaling is not the same as using a multi-layer perceptron though. It seems conceivable that a hidden layer with nonlinearities is the key, so it does seem that a direct comparison to this is needed.

---

> ### Author Response · Authors · 2023-11-23
> **Author Response**
>
> Thank you, we agree that the MLP is technically different than matrix scaling.  We would also ask that you consider [2], wherein an extra set of parameters are used for a domain selector that infers a datapoint's domain and chooses a temperature for scaling.  Comparison is only made to typical temperature scaling.  We believe that our 4 recalibration baselines are very strong.
>
> [2] Yaodong Yu, Stephen Bates, Yi Ma, Michael Jordan. Robust Calibration with Multi-domain Temperature Scaling. https://proceedings.neurips.cc/paper_files/paper/2022/file/b054fadf1ccd80b37d465f6082629934-Paper-Conference.pdf

---

### Author Response · Authors · 2023-11-15
**Author Rebuttal**

We thank all of the reviewers for their detailed and helpful comments.

The main point we would like to make to the reviewers in our rebuttal is the significant difference and accompanying performance gains of our method with respect to the method proposed in [1].  Fisch et al. propose the task of performing selection for the purpose of reducing calibration error.  However, they do not consider recalibration with respect to their selection process.  They only perform recalibration of the pretrained model on the original pretraining distribution, and consider this part of the process for producing the fixed model f.  Then, they use data augmentation to train a (hopefully) robust selector that will be applied to OOD datasets like ImageNet-C and CIFAR-10-C.  However, we contend that possibly the most important step in producing a selectively calibrated classifier is using the selector training data to (jointly) train a recalibrator.

As a result of this difference, the ultimate performance of our algorithms is quite different.  While the results are not directly comparable, we believe that a reasonable comparison can be made using Table 1 in our paper (Section 5.2), where we apply our algorithm to OOD test data, as is done in [1].  On the RxRx dataset, the pretrained model has ECE-2 of 0.331 without using the perturbed validation data to recalibrate.

ECE-2 @ $\beta=1.0$ is lower than 0.331 for every experiment in [1].  In order to reach an ECE-2 of 0.05 in their experiments:
 - CIFAR-10-C: ~60% of data would have to be rejected
 - ImageNet-C: ~75% of data would have to be rejected
 - Lung Cancer: not achieved at any $\beta$

In our RxRx experiment, after recalibration alone, ECE-2 drops to 0.070, and the application of selective recalibration leads to an average ECE-2 of 0.045 in the interval $\beta \in [0.5, 1.0]$.   This performance difference is significant, and highlights that the difference between our algorithms must also be significant with respect to achieving good selective calibration error.

[1] Calibrated Selective Classification. Adam Fisch, Tommi Jaakkola, Regina Barzilay

---

### Meta-Review · Area_Chair_3KX2 · 2023-12-06

**Metareview:**

The paper presents a method for selective recalibration, merging selective classification with calibration, and building on the work of Fisch et al. (2022). It focuses on optimizing rejection and calibration jointly, showing improved calibration over standard methods.

Strengths

- The paper is generally well-written, with clear motivation and includes both empirical and theoretical analyses.
- The concept of joint optimization in selective recalibration is well-conceived, where the contribution is a new loss function, S-TLBCE.

Weaknesses

- The empirical evaluation seems inadequate, lacking in breadth (missing common benchmark datasets) and depth (insufficient dataset and model structure details).
- The paper lacks comparison with more significant calibration and uncertainty approaches, leading to weak baselines.
- Reviewers had questions about the experimental design, particularly the choice of a single epoch for the sequential baseline.
- Classification accuracy can decrease in the results.
- Limited technical novelty and incremental contribution over existing works.

**Justification For Why Not Higher Score:**

See weaknesses

**Justification For Why Not Lower Score:**

N/A

---

### Decision · Program_Chairs · 2024-01-16

Reject